# Molecular Simulations and Drug Discovery of Adenosine Receptors

**DOI:** 10.3390/molecules27072054

**Published:** 2022-03-22

**Authors:** Jinan Wang, Apurba Bhattarai, Hung N. Do, Sana Akhter, Yinglong Miao

**Affiliations:** Center for Computational Biology and Department of Molecular Biosciences, University of Kansas, Lawrence, KS 66047, USA; jawang@ku.edu (J.W.); apurba.bhattarai@ku.edu (A.B.); hungd238@ku.edu (H.N.D.); sana_email@ku.edu (S.A.)

**Keywords:** adenosine receptors, G protein-coupled receptors, mechanisms, molecular simulations, drug discovery

## Abstract

G protein-coupled receptors (GPCRs) represent the largest family of human membrane proteins. Four subtypes of adenosine receptors (ARs), the A_1_AR, A_2A_AR, A_2B_AR and A_3_AR, each with a unique pharmacological profile and distribution within the tissues in the human body, mediate many physiological functions and serve as critical drug targets for treating numerous human diseases including cancer, neuropathic pain, cardiac ischemia, stroke and diabetes. The A_1_AR and A_3_AR preferentially couple to the G_i/o_ proteins, while the A_2A_AR and A_2B_AR prefer coupling to the G_s_ proteins. Adenosine receptors were the first subclass of GPCRs that had experimental structures determined in complex with distinct G proteins. Here, we will review recent studies in molecular simulations and computer-aided drug discovery of the adenosine receptors and also highlight their future research opportunities.

## 1. Introduction

Adenosine (ADO) is an endogenous nucleoside, which regulates multiple biological functions by activating adenosine receptors (ARs) [1,2,3]. ARs belong to the class A G protein-coupled receptors (GPCRs), representing primary targets of approximately 1/3 of marketed drugs [4]. Four subtypes of ARs are expressed in human bodies, including A_1_AR, A_2A_AR, A_2B_AR and A_3_AR. The A_1_AR and A_2A_AR are high-affinity receptors for ADO, whereas A_2B_AR and A_3_AR are low-affinity receptors (Table 1) [5,6].

During function, the A_1_AR preferentially couples to the G_i/o_ proteins to inhibit the production of cAMP by regulating the activity of adenylyl cyclase (AC) [3,7]. It could regulate many cellular responses, such as inhibition of Ca^2+^ conductance and stimulation of phospholipase C, K^+^ conductance, phosphoinositide 3 kinase (PI3K) and mitogen-activated protein kinase (MAPK) [8,9] (Table 1). The A_1_AR is thus considered as a potential drug target for treating myocardial ischemia [10], cardiovascular disorders [11,12], obesity [13] and cancers [14,15]. Particularly, one agonist and four antagonists of A_1_AR were approved in the market (Table 1).

The A_2A_AR prefers to couple to the G_s_ protein to activate AC, resulting in the activation of cAMP-dependent protein kinase A (PKA), protein kinase C (PKC), MAPK and ion channels (Table 1) [16]. The A_2A_AR is widely considered as a potential drug target for treating cardiovascular disorders [11,12], obesity [13], Parkinson’s disease (PD) [17,18] and cancers [14,15]. Particularly, a selective A_2A_AR antagonist Istradefylline was approved for the treatment of PD [19].

The A_2B_AR binds with the G_s_ and G_q_ proteins to induce the PKA signaling to increase the level of cAMP and stimulate phospholipase C and MAPK [20]. The A_2B_AR is recognized as a drug target for treating inflammation [21,22], chronic obstructive pulmonary disease [23] and diabetes [24].

The A_3_AR couples to the G_i_ protein to inhibit AC and decrease cAMP accumulation and PKA activity. Additionally, the A_3_AR could bind with the G_q_ protein to stimulate phospholipase C, resulting in increased Ca^2+^ levels and modulation of PKC activity [25]. In addition, they may activate the phospholipase C pathway through the G_βγ_ subunit [26]. Increasing attention is paid to the A_3_AR for the drug development against inflammation [27], glaucoma [28], rheumatoid arthritis [29] and stroke [30].

Advances of techniques in X-ray crystallography, cryo-electron microscopy (cryo-EM) and Nuclear Magnetic Resonance (NMR) have generated a number of structures for GPCRs including the ARs. They have stimulated numerous structure-based and computer-aided drug design (CADD) developments for ARs [31]. To date (January 2022), Protein Data Bank (PDB) [32] has recorded approximately 64 structures of ARs under different functional states (Table 2). However, there has been no experimental structure for A_2B_AR and A_3_AR yet. Additionally, only static snapshots of ARs could be captured in the PDB structures. It is still challenging for experimental techniques to probe flexibility of the ARs, which plays a critical role in regulating their biological functions and designing potent and selective drug molecules. Molecular dynamics (MD) simulations [33] have proven useful to model protein dynamics and ligand binding/dissociation processes, which has greatly facilitated the rational drug design targeting ARs. Here, we present a review of simulation studies that have revealed important insights into the dynamic and functional mechanisms of ARs, as well as drug discovery efforts targeting the ARs.

## 2. Molecular Simulations Revealed Functional Mechanisms of Adenosine Receptors

### 2.1. Activation of Adenosine Receptors

The classical view of GPCR activation involves conformational states between the active and inactive states [77]. Agonist binding shifts the conformational ensemble of receptor to the activated state to enable coupling of the G protein. Recent successes in structural biology have generated many high resolution structures of ARs under both active and inactive states (Table 2 and Figure 1). Comparison between the active and inactive structures of the A_1_AR [35,38] revealed a large outward movement of transmembrane helix 6 (TM6) intracellular domain during activation of the receptor, which opened the intracellular pocket for G protein binding (Figure 1). Meanwhile, the receptor activation was accompanied by adjustments of the TM7, helix 8 (H8), extracellular loops (ECLs) and ligand binding pocket (Figure 1). A number of important structural motifs, named “microswitches”, play a critical role in the activation of ARs, including the R^3.50^-E^6.30^ salt bridge, the W^6.48^ “rotamer toggle switch” and the Y^7.53^ “tyrosine toggle switch” [78]. The Ballesteros–Weinstein numbering [79] is used for residues in the GPCRs. Although NMR has been used to probe populations of different activation states of the A_2A_AR [58,80,81], it remains challenging to probe dynamic conformational transitions among these different states using experimental techniques. In this regard, MD simulations have been widely applied to study the activation mechanism of ARs.

The R^3.50^-E^6.30^ salt bridge was very stable in MD simulations of the *apo* A_2A_AR in the inactive state [82,83]. Rotameric transition of the W^6.48^ was observed in MD simulations of the agonist 5′-*N*-carboxamidoadenosine (NECA)-bound A_2A_AR, which allowed for water movement through the TM bundle [84]. Such movement further induced a rotational switch of the residue Y^7.53^, inward movement of TM7 on the intracellular side and outward movement of TM6, leading to the opening of the intracellular pocket to facilitate G protein binding [85]. Advances in computing hardware (e.g., GPU and Anton) and software developments have enabled longer conventional MD (cMD) simulations [86]. Even so, cMD is often limited to typically hundreds of nanoseconds to tens of microseconds [87,88]. However, activation of GPCRs including ARs is still beyond the accessible time scale of cMD, which often occurs over milliseconds or even longer timescales [89].

Enhanced sampling techniques that could observe much longer timescale events within shorter simulation time have thus been applied to explore AR activation and deactivation [90,91]. Adaptive sampling combined with Markov State Models (MSMs) allowed for calculation of the activation energy landscape of the *apo* A_2A_AR. The *apo* A_2A_AR sampled four predominant states, including the inactive antagonist bound-like state, the inactive *apo* intermediate state, the agonist-competent state and the active state [92]. Moreover, Li et al. performed extensive Metadynamics simulations to explore the activation and deactivation mechanism of the A_2A_AR [93]. The A_2A_AR was simulated in the agonist-bound, antagonist-bound and *apo* forms. The simulations sampled distinct conformational states of the A_2A_AR that resembled the receptor active and inactive states, involving marked conformational transitions of the W^6.48^ toggle switch. Three distinct regions in the orthostatic binding pocket were identified to be responsible for the ligand binding affinity, selectivity and agonism/antagonism, respectively. In addition, deactivation of the A_1_AR was observed in our recent Gaussian accelerated molecular dynamics (GaMD) simulations of the active receptor after removing the G protein [39]. In summary, MD simulations, especially enhanced sampling, have provided unprecedented insights into the activation mechanism of ARs at the atomistic level.

### 2.2. Specific G Protein Coupling to Adenosine Receptors

The odd ARs (A_1_AR and A_3_AR) preferentially couple to the G_i/o_ proteins, while the even subtypes (A_2A_AR and A_2B_AR) preferentially couple to the G_s_ proteins. However, increasing studies suggest that GPCRs including the ARs could couple to multiple G proteins [94,95,96,97]. MD simulations were applied to investigate the dynamic AR-G protein interactions. The role of G protein in stabilizing the active state of A_2A_AR was investigated by Lee et al. [98]. Four different conformations of A_2A_AR, including the inactive, active-intermediate and fully active in the presence and absence of the mini-G_s_ protein, were used as simulation starting structures. In comparison to the inactive state, lower agonist fluctuations and decreased entropy in the ECLs were observed in the active-intermediate state. In the fully active G protein-bound state, the entropy of ECLs was the highest. The volume of the receptor orthosteric pocket decreased to enable tighter contacts with the agonist. This allowed the G protein-bound active conformation to have better allosteric communication between the G protein binding pocket and the orthosteric site.

In a recent study [99], we employed GaMD simulations on the cryo-EM structures of native agonist ADO-bound A_1_AR-G_i_ and NECA-bound A_2A_AR-G_s_ protein complexes [38,52], as well as “decoy” complexes generated by switching the G proteins (A_1_AR-G_s_ and A_2A_AR-G_i_). GaMD simulations suggested that slight differences of agonist NECA binding in the A_2A_AR were observed upon changing the G_s_ protein to the G_i_ (Figure 2C), while significantly increased fluctuations in the A_1_AR and ADO were identified upon changing the G_i_ protein to the G_s_ (Figure 2D). The agonist ADO sampled two different binding poses (“L1” and “L2”) when the A_1_AR coupled with the G_s_ protein. In the “L2” binding pose, ADO formed interactions with residues Y^1.35^ and Y^7.36^ in the sub-pocket 2 of the A_1_AR as described earlier [35]. Only one stable low-energy conformation was observed for each of the A_1_AR-G_i_ and A_2A_AR-G_s_ complexes as in the cryo-EM structures (Figure 2A,B), being similar for the A_2A_AR-G_i_ complex (Figure 2C). While multiple states of the ADO agonist and G_s_ protein were sampled in the A_1_AR-G_s_ system (Figure 2D). Simulation results thus indicated that the A_1_AR preferred to couple with the G_i_ protein to the G_s_ (Figure 2E), while the A_2A_AR could couple with both the G_s_ and G_i_ proteins (Figure 2F), being highly consistent with experimental findings of the ARs [94,95,96]. Further detailed analysis of the simulation trajectories suggested that remarkably complementary residue interactions at the protein interface led to the specific AR-G protein coupling, which involved mainly the receptor TM6 helix, the G_α_ α5 helix and α4-β6 loop. In summary, the GaMD simulations have provided important insights into the dynamic mechanism of specific GPCR-G protein interactions.

### 2.3. Biased Agonism of Adenosine Receptors

Biased agonism is a phenomenon in which binding of different agonists to a target GPCR promotes distinct receptor conformations that bias cellular signaling toward and away from a subset of pathways, e.g., activation of different G protein or arrestin [100,101,102,103]. Biased agonism was first introduced by Jarpe et al. [103] and has been targeted for developing selective therapeutics of GPCRs [102,104]. The first A_1_AR biased agonist LUF5589 was identified by assessing A_1_AR bias with 800 A_1_AR agonists and antagonists in 2013 [105]. The AR biased agonism has been explored by MD simulations. By combining MD simulations, Gαi/o subunit- and β-arrestin-specific cellular signaling assays, Wall et al. [106] identified that the A_1_AR-selective agonist, BnOCPA, was a biased agonist in exclusively activating the G_ob_ protein. MD simulations were applied on the A_1_AR bound by the biased agonist BnOCPA, neutral agonists ADO and HOCPA and an antagonist PSB36. The simulations suggested that BnOCPA engaged with the same receptor interactions as neutral agonists ADO and HOCPA. However, a distinct partial transition of the N^7.49^PxxY^7.53^ backbone from the active to inactive state was observed in one of the BnOCPA-bound A_1_AR simulations. The α5 helix (GαCT) of the G protein (G_i2_, G_oa_, G_ob_) was dynamically docked to the HOCPA- and BnOCPA-bound active A_1_AR structures to study the agonist-driven interaction between the A_1_AR and the G protein. MD simulations suggested that the GαCT of G_ob_ docked to the A_1_AR via a metastable state (MS1) relative to the canonical state (CS1). The CS1 corresponded to the canonical arrangement as captured in the cryo-EM structure of the A_1_AR-Gi (PDB: 6D9H), whereas the MS1 was similar to the non-canonical state in the neurotensin receptor, being suggested as an intermediate on the way to the canonical state [107]. In contrast, docking of the GαCT of G_oa_ and G_i2_ to the A_1_AR formed MS2 and MS3 states. The MS2 was similar to the β_2_-adrenergic receptor–G_αs_CT complex [108], which was proposed to be an intermediate on the activation pathway and play an important role in G protein specificity. Additional MD simulations were performed on the BnOCPA- and HOCPA-bound A_1_AR in complex with the entire G_oa_ and G_ob_ proteins, respectively. The main differences between the G_oa_ and G_ob_ proteins comprised the formation of transient hydrogen bonds between the α3-β5 and α4-β6 loops of G_oa_ and H8 of the A_1_AR. Overall, G_oa_ interacted more with residues at the TM3 and ICL2 of receptor, while TM5, TM6 and ICL1 were more engaged by G_ob_. Particularly, residues R^7.56^ and I^8.47^ showed a different propensity to interact with G_oa_ or G_ob_ proteins. Therefore, a particular A_1_AR conformation stabilized by BnOCPA may favor different intermediate states during the binding process of G_oa_ and G_ob_ proteins.

In previous studies, bitopic ligands that comprise of orthosteric and allosteric ligand binding moieties were shown to exhibit biased agonism in the A_1_AR [100,109]. The supervised MD (SuMD) simulations were performed to capture the binding of bitopic agonist VCP746 (biased agonist) and its allosteric part (VCP171) to the A_1_AR [110]. The VCP746 sampled the most stable binding pose when the orthosteric site was occupied by the adenosine moiety, suggesting that the agonist component of the VCP746 played a particularly important role in the binding. VCP746 interacted with many receptor side chains and the ECL2 backbone atoms. The linker was captured to insert between the E172^ECL2^ and M^5.35^ side chains in the simulations. The allosteric moiety part VCP171 sampled many orientations on ECL2 but stabilized in a binding mode near ECL2 when the adenosine moiety reached the orthosteric binding site. Side chains E172^ECL2^ and K173^ECL2^ formed a sort of saddle for the allosteric moiety, which often oriented the 3-(trifluoromethyl)phenyl group toward the hydrophobic pocket formed by residues K173^ECL2^ and I167^ECL2^. In comparison, the positive allosteric modulator (PAM) VCP171 usually oriented its 3-(trifluoromethyl)phenyl moiety toward the top of ECL2. Taken together, these computational findings suggested a different binding mode for VCP171 as part of VCP746 and proposed an allosteric site of ECL2 as involved in the observed bias (Figure 3).

### 2.4. Allosteric Modulation of Adenosine Receptors

Due to high similarity of the orthosteric site among different ARs, agonists have failed clinical trials due to off-target side effects. To overcome this problem, allosteric modulators have been developed that bind topographically distinct “allosteric” sites in the receptor (Figure 3) and regulate the effects of orthosteric ligands. PAMs potentiate the effects of orthosteric ligands, whereas negative allosteric modulators (NAMs) do the opposite. AR allosteric modulators provide subtype selectivity reducing the off-target side effects and hence represent a promising approach to selective drug design. Gao et al. [111] characterized the binding and functional antagonism of fluorescent conjugates of xanthine amine congener (XAC) and SCH442416 to the A_2A_AR. Among antagonists tested, MRS7322, MRS7396, MRS7416, XAC245 and XAC630 behaved as allosteric antagonists of A_2A_AR, whilst MRS7395 and XAC488 acted as competitive antagonists [111]. Allosteric antagonists MRS7396 and 5-(*N*,*N*-hexamethylene)amiloride (HMA) were more potent than MRS7416 in displacing [^3^H]ZM241385 binding, while MRS7396, XAC630 and HMA were less potent than radioligand binding in displacing MRS7416 binding [111]. Mutation of D52N in the sodium site of A_2A_AR changed the affinity of HMA and MRS7396, indicating preferences for different A_2A_AR conformations [111].

In one study, we performed GaMD simulations to identify binding modes of two prototypical PAMs in the A_1_AR [112]. VCP171 and PD81723 were initially placed around 20 Å away from the receptor. Amber [113] and NAMD [114] software packages were used to perform enhanced sampling of PAMs binding to the receptor. GaMD predicted the binding modes of both the PAMs near ECL2 site which was consistent with the experimental mutagenesis results [115]. In the PAM-bound state of the receptor, the NECA agonist exhibited lower fluctuations in the orthosteric site. In contrast, without the PAM, the agonist was observed to be very dynamic in the orthosteric site and could even dissociate in some of the simulations.

Very recently, using the first cryo-EM structure of A_1_AR bound to a PAM MIPS521 in presence of bound ADO agonist and G protein, we performed further GaMD simulations to characterize molecular basis of allosteric modulation [39]. GaMD simulations showed that the MIPS521 PAM molecule could stabilize the ADO in the orthosteric pocket (Figure 4). Even in the absence of G protein, the PAM molecule could show positive cooperativity stabilizing the ADO agonist. We could observe deactivation during GaMD simulation of A_1_AR without the PAM and G protein, whereas the presence of MIP521 could slow deactivation of A_1_AR without the G protein bound (Figure 4). This showed that the MIP521 PAM could stabilize the ADO-bound of A_1_AR in a “G-protein-bound-like” conformation.

SuMD simulations were also performed to investigate the binding of LUF6000 PAM to the A_3_AR [116]. The SuMD simulations characterized the binding pathway of LUF6000 PAM to the A_3_AR in presence of agonist ADO. In presence of the agonist, LUF6000 bound to the receptor in two different poses. First, the PAM bound to the ECL2 site changing its conformation which further induced energetically favorable agonist interactions in the orthosteric site. Second, the PAM stably bound near the mouth of the orthosteric site stabilizing the ternary complex with agonist-bound receptor. In another study, SuMD simulations were performed to study the allosteric role of sodium ion binding to the A_2A_AR [117]. The sodium ion has been proposed as a NAM as it was observed in a distal site compared to the orthosteric site [71]. The sodium ion was able to coordinate the inactive A_2A_AR without any conformational changes in the SuMD simulations. However, the TM helices rearranged to accommodate the sodium ion in an intermediate-active state of A_2A_AR. Deganutti et al. performed SuMD simulations to capture binding of the antagonist 13B, bitopic agonist VCP746 and PAMs PD81723 and VCP171 to the A_1_AR [110]. The SuMD simulations showed that PAMs can bind to several receptor sites rather than a single allosteric pocket. In absence of the NECA agonist, PAM showed partial agonism behavior. Despite having structural similarities between PAMs, different binding paths were observed in the simulations, which revealed dramatic effects of subtle chemical modifications in the ligand structure.

### 2.5. Ligand Binding to Adenosine Receptors

Ligand binding kinetics, especially the dissociated rate, have recently been recognized to be potentially more relevant for drug design. MD simulations have been performed in order to understand ligand binding/unbinding of ARs. In a study combining MD simulations and kinetic radioligand binding experiments, Guo et al. investigated the dissociation mechanism of an antagonist ZM241385 to the A_2A_AR [118]. MD simulations that captured the dissociation of the antagonist ZM241385 from the A_2A_AR helped identify the residues in the ligand unbinding pathway. Experiments validated that mutation of these residues could influence ligand’s dissociation rate dramatically even though the binding affinity was barely changed. This study demonstrates that receptor structural elements that are not important to binding affinity can prove key to ligand kinetics.

SuMD simulations were performed to study the binding interactions of ADO and its metabolite inosine ligand to the A_2_AR in both the active-intermediate and its G protein-bound conformations [119]. During the SuMD simulations of ADO-bound A_2A_AR, the ligand was stabilized by two hydrogen bonds formed with residues N^6.55^ and E169^ECL2^ in the orthosteric pocket. Conversely, inosine could form only one hydrogen bond with N^6.55^. Interestingly, ligand binding in the orthosteric pocket of both systems was remarkably influenced by the presence of G protein. In another SuMD study, Sabbadin et al. explored the recognition pathway of ADO by the A_2A_AR [120]. SuMD simulations showed that ECL3 represents a possible metastable binding site for ADO. During the binding process, ECL3 helped orient the adenosine ribose ring toward orthosteric entrance. In the orthosteric binding site, ribose moiety of the ligand experienced dynamic flipping between “ribose-down” and “ribose-up” conformations. Bolcato et al. performed SuMD simulations to study binding of subtype selective antagonists LC4 and Z48 binding to A_1_AR and A_2A_AR, respectively [121]. The simulations showed that receptor-ligand recognitions were multistep processes involving intermediate states to guide the (un)binding events. Overall, Z48 favored A_2A_AR over A_1_AR, forming classic antagonist fingerprint interactions at the orthosteric site. In case of the A_1_AR, a water molecule was seen playing a key role in stabilizing LC4 at the orthosteric pocket, which was not observed in the case of A_2A_AR. In another study, Deganutti et al. [122] performed SuMD simulations to investigate the binding/unbinding pathways of five different A_1_AR agonists including ADO, CPA, NECA, HOCPA and BnOCPA. The SuMD simulations showed that the ligand followed the binding paths involving mainly ECL2, the top part of TM1, TM2, TM6 and TM7. The ligand dissociated following similar paths; however, ECL2 was less engaged. These pathways were further supported by alanine mutagenesis experiments.

Recently, we performed GaMD simulations to determine the binding and dissociation pathways of caffeine (CFF) antagonist to the A_2_AR [123]. The X-ray structure of A_2A_AR in complex with CFF (PDB: 5MZP) [36] was used as initial conformation. A total of 10 CFF molecules were placed randomly at a distance >15 Å from the extracellular surface of the A_2A_AR. Spontaneous binding and dissociation of CFF in the receptor were successfully captured in the GaMD simulations. A main binding pathway of CFF to the A_2A_AR was identified from the 63-ns GaMD equilibration (Figure 5A). CFF reached its binding site of the A_2A_AR through interacting with ECL2, ECL3, TM7 and finally the receptor orthosteric site (Figure 5D). GaMD production simulations captured a slightly different binding pathway when the orthosteric pocket was already occupied by one CFF molecule (Figure 5B). The second CFF explored a region between ECL3 and TM7 during this binding process (Figure 5E). The dissociation pathway of CFF was mostly the reverse of the dominant binding pathway (Figure 5C,F).

### 2.6. Lipid Interactions with Adenosine Receptors

Lipid bilayers have been shown to modulate GPCR functions, including conformation stability, ligand binding and oligomerization [124,125,126]. Modulatory effects mediated via changes in the physical properties of membrane, such as thickness, curvature and surface tension, have been extensively studied using experimental and computational methods [126,127]. Here, we discuss MD studies focusing on AR-lipid interactions.

In one study, GaMD simulations were applied to study the relationship between the lipid environment and A_1_AR activation state [128]. The cryo-EM structure of the active ADO-bound A_1_AR coupled with the G_i_ protein (A_1_AR-Gi) [38] and the X-ray structure [35,36] of the inactive antagonist PSB36-bound A_1_AR (PSB36-A_1_AR) were used to perform GaMD simulations. The A_1_AR was embedded in a 1-palmitoyl-2-oleoyl-glycero-3-phosphocholine (POPC) lipid bilayer. GaMD simulations suggested that the membrane lipids play a critical role in stabilizing different states of the A_1_AR. Different structural flexibility profiles were identified in GaMD simulations of the inactive and active A_1_AR. Compared with the inactive A_1_AR, higher fluctuations were found at the ECL2 region, intracellular ends of TM5 and TM6 in the active A_1_AR. The receptor TM domain and the ligands were rather rigid. However, the G protein coupled to the active A_1_AR exhibited high flexibility, especially in the, α4-β5 loop and α4-β6 loop and α5 helix of the Gα subunit and terminal ends of the G_βγ_ subunits.

The -S_CD_ order parameter values obtained from GaMD simulations agreed well with experimental data. In NMR experiments, the -S_CD_ for the fifth carbon C-H bond of POPC was at ~0.18–0.20 [129]. The -S_CD_ value of POPC’s fifth carbon atom was ~0.17 ± 0.02 in the lower leaflet in the active A_1_AR system. It increased to ~0.20 ± 0.02 in the inactive A_1_AR system (Figure 6). The -S_CD_ value of the ninth carbon C-H bond in POPC calculated from GaMD simulations was ~0.10, being consistent with the NMR experiments [129]. Additionally, the GaMD simulations suggested that POPC lipids in the lower leaflet of the inactive A_1_AR system were less fluid than in the active A_1_AR system. The similar -S_CD_ values of sn-2 acyl chains of POPC molecules in the upper leaflet were identified in the inactive and active A_1_AR systems. However, the -S_CD_ for the lower leaflet in the inactive A_1_AR system was larger than those in the active A_1_AR system. This finding correlated well with the TM6 outward movement in the active A_1_AR, which caused higher inclination of the C-H bonds to be aligned along the bilayer normal. In summary, GaMD simulations have revealed strongly coupled dynamics between a GPCR and the membrane lipids that depend on the receptor activation state.

MD simulations were also performed to explore the effects of different lipids in antagonist caffeine binding to A_2A_AR [130]. Only POPC, a mixture of POPC and 1-palmitoyl-2-oleoyl-sn-glycero-3-phosphatidylethanolamine (POPE), and cholesterol rich lipid environment were used in the study. Simulations showed that H8 folding depended on the lipid environment. Cholesterol was observed to bind a cleft between TM1 and TM2, which stabilized a distinct caffeine binding pose. This further highlighted the importance of using physiological cholesterol concentration in MD simulations. Recently, Bruzzese et al. performed MD simulations to study the role of different lipids in activation of A_2A_AR [131]. Inactive A_2A_AR in the *apo* or agonist (ADO or NECA) bound conformations were used as starting structures in either 1,2-dioleoyl-sn-glycerol-3-phosphoglycerol (DOPG) or 1,2-dioleoyl-sn-glycerol-3-phosphocholine (DOPC) lipid bilayer. DOPC could facilitate the transition of the inactive A_2A_AR to an intermediate conformation. In DOPG with *apo* conformation of A_2A_AR, the receptor could also transit to an intermediate conformation. Interestingly, the agonist-bound A_2A_AR transitioned into a fully active state. These variations were attributed to the allosteric effects mediated by the lipid and presence/absence of an agonist. Leonard et al. performed 35 µs MD simulations to study the preference for lipid solvation in the A_2A_AR [132]. Free energy of lipid solvation was calculated for different activation states of A_2A_AR with different lipids. The results showed that the inactive state preferred unsaturated lipids over the saturated ones for formation of the first solvation lipid shell around the receptor. This preference enhanced even more in the partially active A_2A_AR state as compared to that of the inactive receptor.

In addition to the above-mentioned all-atom MD simulations, coarse-grained models were also widely used to reach longer time scale and/or simulate larger systems. For example, both all-atom and coarse-grained MD simulations were performed to study the effects of membrane lipids and cations in the inactive, intermediate and fully active G protein-bound conformations of A_2A_AR [133]. The study was performed in both detergent micelles and lipid bilayer environment. MD simulations suggested that phosphatidylinositol bisphosphate 2 (PIP2) interacted with the A_2A_AR intracellular residues, which could reduce the flexibility of the receptor in the inactive state and limit the transition to the active-intermediate state. For the fully active A_2A_AR, PIP2 stabilized the receptor–G protein complex. However, such stabilizing interactions were absent in the non-ionic micelles. The level of activation microswitches observed in the POPC lipid bilayer and detergent micelles were different, suggesting a rheostat model of GPCR activation microswitches as compared to a binary switch model. Song et al. [134] used coarse-grained MD simulations to study A_2A_AR-lipid interactions. Simulations showed that different kinds of lipids could interact with nine binding sites on the receptor. The lipids were observed to allosterically modulate activation of A_2A_AR. PIP2 could help stabilize the active state of the A_2A_AR by stabilizing outward movement of TM6 and enhancing the AR-G protein interactions. These studies strengthened the notion that lipids affect GPCR signaling and function by allosterically modulating its membrane lipid environment, being consistent with experimental findings [135]. For example, Huang et al. [135] identified cholesterol as a weak PAM of the A_2A_AR by combining ^9^F NMR, computational analysis and G protein assays. Their GTP hydrolysis assays showed a marginal increase in basal activity with increasing cholesterol, in addition to a weak enhancement in the agonist potency. Furthermore, their ^19^F NMR data suggested that the enhancement resulted from an increase in the receptor’s active state population and a G protein-bound pre-coupled state.

## 3. Computer-Aided Drug Discovery of Adenosine Receptors

More than 515 million compounds are available in the ZINC chemical database [136]. It is impossible to perform high-throughput screening of all the chemical compounds. In this regard, virtual screening is a valuable strategy to efficiently screen a large number of compounds and select only a small subset of promising compounds for testing in in vitro and in vivo experiments [31,137,138]. Rodríguez et al. [139] identified 20 potent ligands by molecular docking from a pool of more than 6.7 million commercially available compounds [139]. MD simulations have been incorporated into virtual screening to increase its success rate by providing a more reliable receptor structural ensemble and more accurate binding free energy calculation. In the following, we discuss novel binding site identification and drug design of ARs.

### 3.1. Drug Binding Sites of Adenosine Receptors

The orthosteric site has long been targeted to design drugs for ARs. However, this site is usually conserved across four subtypes of ARs. Consequently, it is critical to identify novel, less conserved binding sites to design selective ligands of ARs. Caliman et al. [140] used FTPMap to identify potential binding sites of the A_2A_AR in the 20 A_2A_AR crystal structures and 30 receptor clusters with 1.57 μs and 1.75 μs MD simulations of the *apo* receptor generated from structures of 3EML and 3QAK. Five “non-orthosteric” sites were identified, including the extracellular region of TM3 and TM4, lipid interface of TM5 and TM6, intracellular region between TM1 and TM7, G protein binding site among TM2, TM3, TM6 and TM7 and intracellular region of TM3 and TM4 [140].

Another method, the site-identification by ligand competitive saturation (SILCS) [141,142] method, was also developed to predict the location and approximate affinities of small molecular fragments on a target receptor surface by performing MD simulations. Moreover, MD simulations and mutation experiments were combined to identify residue hotspots of ARs for ligand binding. Wang et al. [143] explored the importance of specific residue hotspots by mutating Leu^6.51^ to Val specific to the A_2B_AR in the A_2A_AR. They incorporated MD simulations and radioligand binding assays to validate the mutation. A selective A_2A_AR antagonist ZM241385 indicated decreased affinity for the L249V^6.51^ mutant of A_2A_AR, suggesting the important role of this Leucine residue.

### 3.2. Binding Free Energy Calculations of Adenosine Receptors

One of the fundamental assumptions of ligand-based drug design is that similar molecules will have similar biological activity. Such an approach often fails when a small change in the ligand structure leads to a drastic difference in binding affinity [144]. In this regard, MD simulation and binding free energy calculation methods, including molecular mechanics Poisson–Boltzmann surface area (MM/PBSA), molecular mechanics generalized Born surface area (MM/GBSA) and free energy perturbation (FEP), have become valuable tools to predict ligand binding affinities for drug design [145,146]. MM/PBSA and MM/GBSA are popular methods for binding free energy prediction in drug design since they are more accurate than most scoring functions of molecular docking [146]. For example, Lenselink et al. [147] identified two ligands of A_2A_AR by applying MM/GBSA to rescore binding poses from Glide docking. Compared with MM/PBSA or MM/GBSA, MD/FEP is a more accurate binding free energy calculation method. Therefore, MD/FEP has been routinely used in drug design projects [45,145,148,149,150,151,152]. For the A_2A_AR, MD/FEP calculations suggested the loss of binding of 3-deazaadenosine due to modification of a single heavy atom in ADO [150]. Even for such a small modification, MD/FEP was able to successfully predict the change of ligand binding affinity. One lead compound predicted from MD/FEP was synthesized and experimentally verified to be a full agonist equipotent to ADO. Furthermore, MD/FEP suggested that water in the binding site provided a major driving force for the ligand–protein interaction.

Fragment-based drug discovery relies on successful optimization of weak ligands for binding affinity and selectivity. MD/FEP has been widely applied in the fragment-to-lead optimization process. Matricon et al. [149] discovered that the benzothiazole fragment bound to the A_1_AR through hydrogen bond interactions with N^6.55^. Through a single vector growth, they designed nine additional compounds from the initial fragment, all of which showed improved binding affinity as suggested by MD/FEP calculations [149]. Eventually, the compounds were synthesized and their binding affinities in the A_1_AR were experimentally measured by radioligand binding assays [149]. The resulting ligands led to >1000-fold improvements of binding affinity and nearly 40-fold higher subtype selectivity.

Furthermore, MD/FEP has been used to identify the ligand binding pose. In a proof-of-concept study, Jespers et al. [45] presented a robust protocol based on iterations of MD/FEP calculations, chemical synthesis, biophysical mapping and structural biology to determine the binding pose of a series of antagonists in the A_2A_AR. Eight binding site mutations in the A_2A_AR were initially analyzed with MD/FEP calculations. The suggested ligand binding pose was subsequently used to guide the design of the new analogues. Remarkably, the binding affinities obtained from MD/FEP prediction were highly consistent with the experimental data. Furthermore, the predicted binding poses were highly consistent with the experimental structures.

### 3.3. Design of Biased Agonists of Adenosine Receptors

Wall et al. [106] discovered biased agonist BnOCPA of A_1_AR that favors G_ob_ over other G protein subtypes and hence produces analgesic effects without sedation, bradycardia and hypotension. MD simulations revealed the BnOCPA binding modes in the receptor. BnOCPA binding resulted in unique active- and inactive-like conformations of the receptor during the simulations. Similarly, Baltos et al. [153] investigated the A_3_AR biased agonists and discovered that several methyluronamide nucleoside derivatives exhibited biased agonism. In particular, molecular docking of MRS5679, an (*N*)-methanocarba nucleoside derivative with an extended C2 substituent, to a homology model of the A_3_AR provided important insights into the molecular mechanism of biased signaling. Binding of the MRS5679 biased agonist favored a distinct conformation of the A_3_AR with outward movement of the TM2 extracellular domain. Valant et al. [109] combined the endogenous agonist and allosteric modulator to design a bitopic ligand VCP746 as a biased agonist of A_1_AR that favored cAMP pathway over ERK1/2 phosphorylation pathway. Molecular docking calculations and SuMD simulations [110,154] were used to study ligand binding to the receptor (see Section 3.4).

### 3.4. Design of Allosteric Modulators of Adenosine Receptors

Virtual screening has been widely used for agonist/antagonist design targeting GPCRs [31]. However, it is rather challenging to apply virtual screening to discover allosteric modulators with high potency. This is largely due to the lower affinity of allosteric modulators compared with the agonists/antagonists and high receptor flexibility of the target sites. Induced-fit docking [155] and ensemble docking [156] have been developed to account for receptor flexibility in virtual screening. The structural ensembles of target receptors could be taken from NMR structures, homology modeling, MD simulations or any other conformational sampling method [156,157,158,159]. It has been applied in AR allosteric modulator design. The receptor structural ensemble is often generated from MD simulations. Particularly, enhanced sampling MD simulations could sample larger conformational space of the drug target and have thus been shown to increase the success rate of finding allosteric modulators of the M_2_ muscarinic GPCR [160]. In a recent study [161], we tested whether the receptor ensemble generated from GaMD simulations could increase the docking accuracy of discovering PAMs in the A_1_AR. Extensive retrospective ensemble docking calculations of PAM binding to the A_1_AR were performed using GaMD simulations and *AutoDock* (Figure 7) [162]. The dihedral and dual boost GaMD implemented in the AMBER and NAMD were performed. The rigid-body docking at short, medium and long levels and flexible docking were all evaluated in the ensemble docking protocol. The boost potential obtained for the same system with AMBER was larger than with NAMD in the GaMD simulations, which is due to different algorithms used to calculate the system potential statistics [112]. Accordingly, larger conformation space of the receptor was sampled in the AMBER simulations, leading to improved docking performance. Correction of docking score by the GaMD reweighted free energy of the receptor structural cluster further improved the docking performance. With GaMD reweighted scores, ranking by the average binding energy ($BE_{avg}$) performed better than by the minimum binding energy ($BE_{min}$) in terms of the area under the receiver operating characteristic curves (AUC), enrichment factors (EFs) metrics. The receptor ensemble obtained from AMBER dual boost GaMD simulations of the VCP171 PAM-bound ADO-A_1_AR-Gi outperformed other receptor ensembles for docking. This ensemble consisted of conformations of the *holo* A_1_AR with PAM bound at the ECL2 allosteric site. Interactions between the PAM and receptor ECL2 might induce more reliable conformations for PAM binding, which were otherwise difficult to sample in the simulations of PAM-free (*apo*) A_1_AR. Ensembles obtained from dual boost GaMD performed better than the dihedral-boost GaMD for docking, suggesting that GaMD with a higher acceleration level was needed to sufficiently sample conformational space of the GPCR PAM binding site.

Overall, flexible docking performed significantly better than the rigid-body docking at different levels with *AutoDock*. This suggested that the flexibility of protein side chains in ensemble docking also played an important role. The side chains of representative receptor structures obtained from GaMD simulations might be still in unfavored conformations for PAM binding. Flexible docking of protein side chains could then alleviate this problem to achieve better performance. In summary, GaMD simulations and flexible docking greatly improved the docking performance by effectively accounting for the protein flexibility in both the backbone and side chains. Such an ensemble docking protocol will greatly facilitate future allosteric drug design of the A_1_AR.

### 3.5. Design of Bitopic Ligands of Adenosine Receptors

Bitopic ligands contain hybrid molecular structures with pharmacophores of both orthosteric and allosteric ligands. Bitopic drug design has gained popularity because of its multivariate functions including increased efficacy and biased agonism in one ligand. Narlawar et al. [163] designed bitopic ligands of A_1_AR combining orthosteric and allosteric pharmacophores with increasing length of linker between them. In particular, LUF6258 with a 9-carbon atom spacer between the allosteric and orthosteric moieties stood out with increased efficacy. The homology model of the A_1_AR was used for docking of compounds and the binding poses were analyzed. LUF6258 had its adenosine part in the orthosteric pocket, whereas the allosteric moiety could extend out to the extracellular space and interacted with the ECL2 region. Valant et al. [109] rationally designed a new bitopic ligand VCP746 by combining endogenous agonist adenosine and PAM VCP171. VCP746, in an A_1_AR-mediated inhibition assay, showed 30-fold biased agonism toward cAMP pathway in comparison to ERK1/2 phosphorylation pathway. In a follow up study [154], structural changes were made to VCP746, and structural–activity relationship experiments suggested that the 4-(trifluoromethyl) phenyl allosteric pharmacophore group was responsible for the ligand bias effect in A_1_AR. Docking calculations using the A_1_AR homology model suggested that the allosteric and linker region of the bitopic ligand explored the extracellular vestibule of the receptor. Deganutti et al. [110] applied SuMD to simulate the binding of VCP746 to A_1_AR and proposed the binding modes of the bitopic ligand. The adenosine moiety of the ligand bound stably at the orthosteric site, whereas the VCP171 part bound near the ECL2 region. These molecular details further helped understanding the functional mechanism of the bitopic ligand. These studies provide evidence that combining orthosteric and allosteric pharmacophores improves pharmacological drug properties such as on-target efficacy, biased agonism, less on-target side effects, etc.

## 4. Conclusions

ARs have served as established drug targets. Computer-aided structure-based drug design approach has proven useful as while traditional agonists and antagonists often induce adverse side effects, new paradigms have emerged to search for novel biased agonists and allosteric modulators that can serve as selective drug molecules of the ARs. In this regard, because the target sites of these new ligands often involve regions on the receptor surface, it is crucial to account for receptor flexibility in order to design potent drug molecules. Additionally, kinetics of ligand binding has recently been recognized to be potentially more relevant for drug design. In particular, the dissociation rate constant that determines the drug residence time appears to better correlate with drug efficacy than the binding free energy [164,165]. However, ligand kinetic rates are even more difficult to compute than the binding free energies, largely due to slow processes of ligand binding and dissociation over long time scales [165]. Remarkable advances in MD approaches and computer hardware have paved the way to capture the ligand binding/unbinding processes of ARs in molecular details [166], which is expected to play a more important role in drug design in the future. In this context, advanced computational platforms (e.g., ANTON 3 [167] and Deep Docking [168]) and improved simulation techniques, including the coarse-grained MD, FEP, ligand and peptide GaMD [169,170], SuMD [110], Metadynamics [93], machine learning [171] and deep learning [168,172,173], will continue to drive rational computer-aided design of more potent and selective drug molecules of ARs.


## Figures and Tables

**Figure 1 molecules-27-02054-f001:**
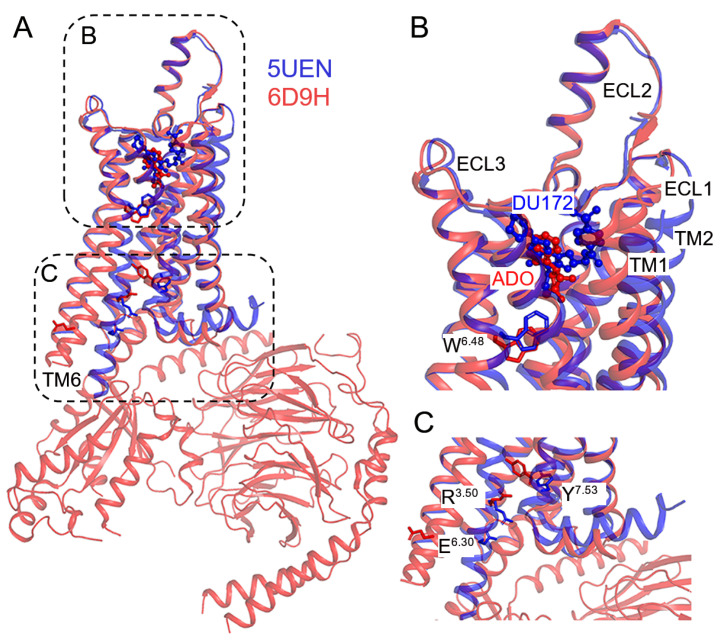
Comparison of experimental structures of the inactive antagonist-bound (PDB: 5UEN) and active agonist-G protein-bound (PDB: 6D9H) A_1_AR. (**A**) Outward movement of transmembrane helix 6 (TM6) in the active agonist-bound receptor (red) induces opening of the intracellular pocket for binding with G protein as compared to the inactive antagonist-bound conformation of the receptor (blue). (**B**) The receptor activation is accompanied by adjustments of the ligand binding pocket, extracellular loops (ECLs) and the W^6.48^ “rotamer toggle switch”. Antagonist DU172 (blue) and agonist adenosine (ADO, red) occupy different regions of the orthosteric pocket in the inactive and active conformations of the A_1_AR receptor, respectively. (**C**) “Microswitches” play a critical role in the activation of adenosine receptors, including the R^3.50^-E^6.30^ salt bridge and the Y^7.53^ “tyrosine toggle switch”.

**Figure 2 molecules-27-02054-f002:**
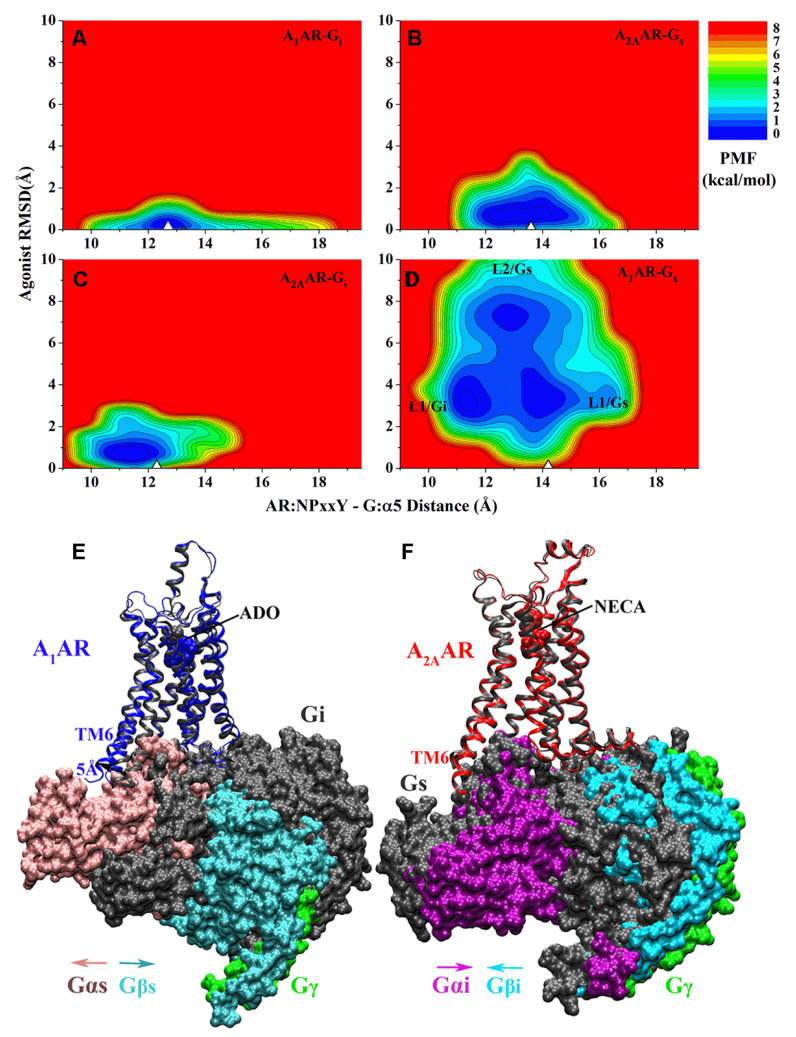
GaMD simulations revealed mechanism of specific G protein coupling to adenosine receptors: 2D potential of mean force (PMF) profiles of the (**A**) A_1_AR-G_i_, (**B**) A_2A_AR-G_s_, (**C**) A_2A_AR-G_i_ and (**D**) A_1_AR-G_s_ complex systems regarding the agonist RMSD relative to the cryo-EM conformation and AR:NPxxY-G:α5 distance. The white triangles indicate the cryo-EM or simulation starting structures. Summary of specific AR-G protein interactions: (**E**) the ADO-bound A_1_AR prefers to bind the G_i_ protein to the G_s_. The latter could not stabilize agonist ADO binding in the A_1_AR and tended to dissociate from the receptor. (**F**) The A_2A_AR could bind both the G_s_ and G_i_ proteins, which adopted distinct conformations in the complexes. Adapted from reference [99] with permission from American Chemical Society. Further permissions related to the material excerpted should be directed to the American Chemical Society.

**Figure 3 molecules-27-02054-f003:**
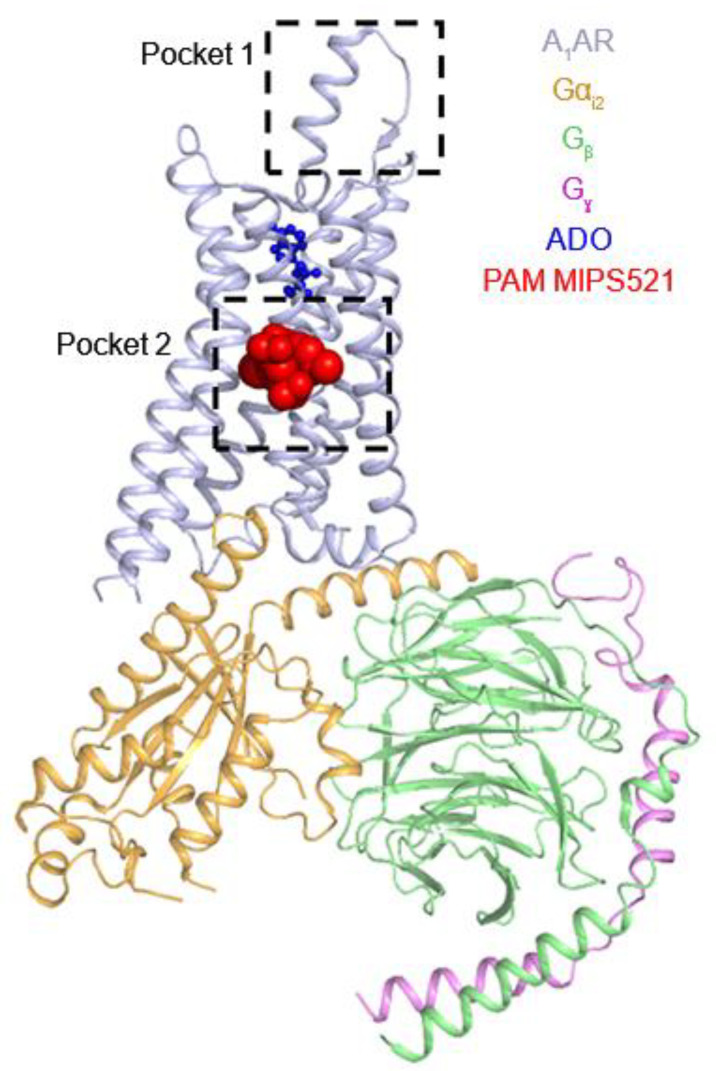
The “allosteric” binding sites (pocket 1 and pocket 2) in the A_1_AR. The cryo-EM structure of A_1_AR–Gi2 complex bound by the PAM MIPS521 (pocket 2) was shown. Another allosteric binding site (pocket 1) was suggested by mutation experiments and MD simulations.

**Figure 4 molecules-27-02054-f004:**
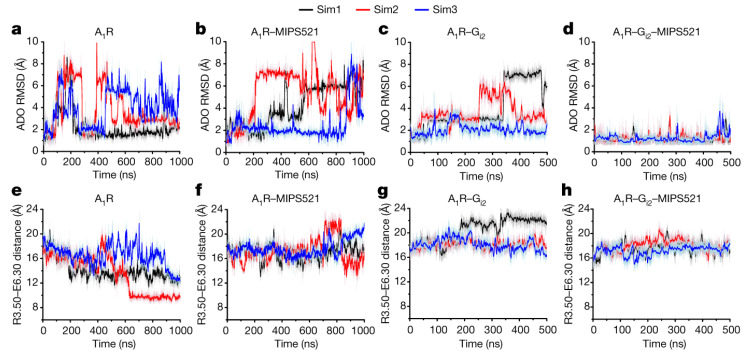
MIPS521 PAM molecule stabilized the ADO agonist binding and the A_1_AR-G_i2_ protein complex, as well as slowing deactivation of A_1_AR upon removal of the G_i2_ protein. RMSD (Å) of ADO from GaMD simulations completed in the absence (**a**) or presence (**b**) of MIPS521, G_i2_ (**c**) or both (**d**). (**e**–**h**) Distance between the intracellular ends of TM3 and TM6 (measured as the distance in Å between charge centers of residues R^3.50^ and E^6.30^) in the absence (**e**) or presence (**f**) of MIPS521, G_i2_ (**g**) or both (**h**). Each condition represents three GaMD simulations, with each simulation trace displayed in a different color (black, red, blue). The lines depict the running average over 2 ns. Reprinted from reference [39] with permission from Springer Nature.

**Figure 5 molecules-27-02054-f005:**
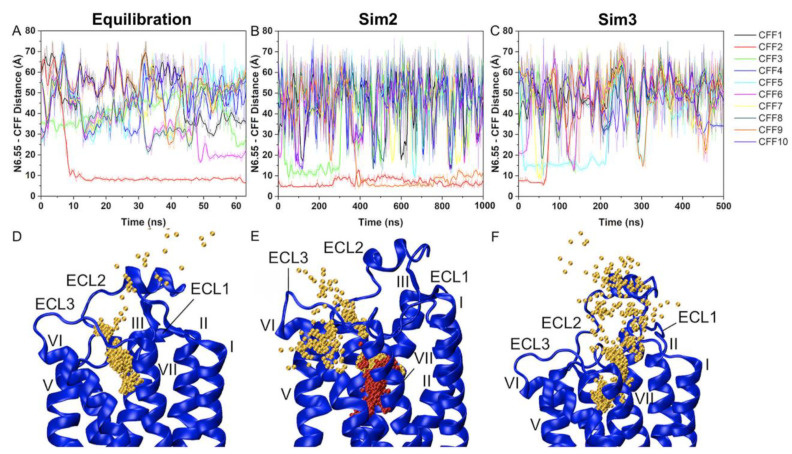
Binding and dissociation pathways of caffeine (CFF) from the A_2A_AR were revealed from GaMD simulations. (**A**–**C**) Time courses of the N6.55:ND2-CFF:N1 distance calculated from GaMD equilibration, Sim2 and Sim3 GaMD production simulations. (**D**–**F**) Trace of CFF (orange and red) in the A_2A_AR observed in the GaMD equilibration, Sim2 and Sim3 GaMD production simulations. The seven transmembrane helices are labeled I to VII, and extracellular loops 1–3 are labeled ECL1–ECL3. Reprinted from reference [123] with permission from Frontiers.

**Figure 6 molecules-27-02054-f006:**
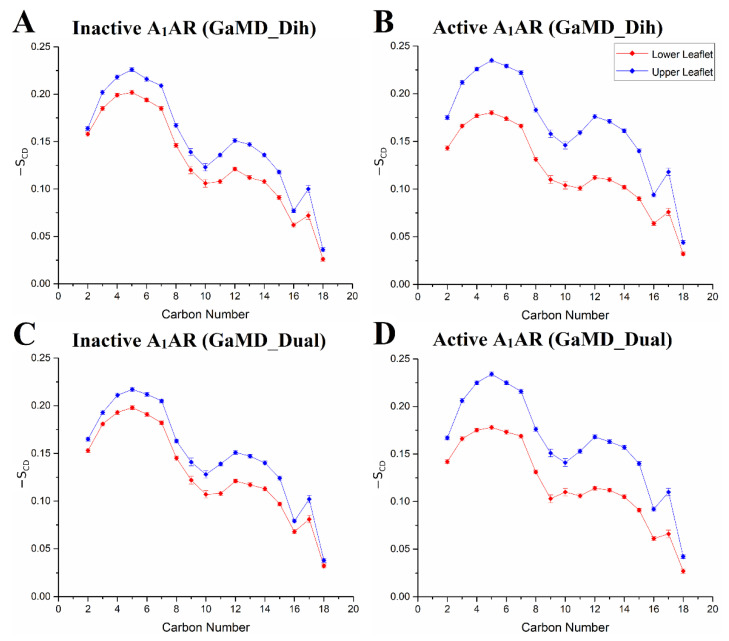
Differentiated order parameters of lipid molecules were found in simulation systems of the inactive and active A_1_AR: (**A**) inactive A_1_AR using dihedral-boost GaMD, (**B**) active A_1_AR using dihedral-boost GaMD, (**C**) inactive A_1_AR using dual-boost GaMD and (**D**) active A_1_AR using dual-boost GaMD. Red diamond lines represent the average -S_CD_ order parameters for the cytoplasmic lower leaflet and blue diamond lines for the extracellular upper leaflet. Reprinted from reference [128] with permission from Wiley.

**Figure 7 molecules-27-02054-f007:**
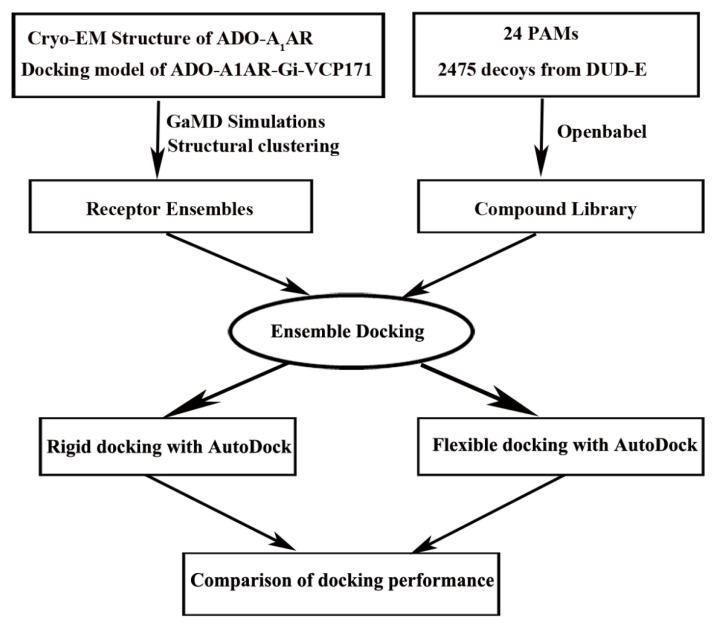
Overview flowchart for retrospective docking of positive allosteric modulators (PAMs) in the A_1_AR: Starting from the cryo-EM structure of the active ADO-Gi-bound A_1_AR (6D9H) and docking model of PAM VCP171-bound A_1_AR (ADO-A_1_AR-Gi-VCP171), GaMD simulations were carried out to construct structural ensembles to account for the receptor flexibility. Meanwhile, a compound library was prepared for 25 known PAMs of the A_1_AR and 2475 decoys obtained from the DUD-E with *openbabel* 2.4.1. Ensemble docking was then performed to identify the PAMs for which the AUC and enrichment factors were calculated to evaluate docking performance. Both rigid-body and flexible docking were tested using *AutoDock*. Reprinted from reference [161] with permission from Elsevier.

**Table 1 molecules-27-02054-t001:** Signaling and approved drugs of adenosine receptors.

**Name**	A_1_AR		A_2A_AR		A_2B_AR	A_3_AR
**G protein coupling**	G_i_, G_o_		G_s_		G_s_, Gq/11	G_o_, G_q/11_
**Downstream signaling**	↓AC		↑AC		↑AC	↓AC
	↑Phospholipase C		↑MAP Kinase		↑Phospholipase C	↑Phospholipase C
	↑K^+^ channel, ↓Ca^2+^ channel		↑PKA		↑PKA	↑Ca^2+^ channel
	↑PI3K		↑PKC		↑MAP Kinase	↑PI3K
	↑MAP Kinase					↑PKC
						↑MAP kinase
**Adenosine binding affinity**	5.10 nM		30.9 nM		1000 nM	100 nM
**Approved drugs**	Drug	Therapeutic use	Drug	Therapeutic use		
	Adenosine (agonist)	Paroxysmal supraventricular tachycardia	Adenosine (agonist)	Myocardial perfusion imaging		
	Regadenoson (antagonist)	Asthma	Istradefylline (antagonist)	Parkinson’s disease		
	Theophylline (antagonist)	Asthma				
	Doxofylline (antagonist)	Asthma				
	Bamifylline (antagonist)	Asthma				

**Table 2 molecules-27-02054-t002:** The available experimental structures of the A_1_AR and A_2A_AR deposited in the Protein Data Bank (PDB) along with their PDB accession codes, conformational states (inactive, partially active or fully active), the functional effect (agonist or antagonist) of the co-crystallized ligand.

PDB Code	Resolution (Å)	Binding Affinity	Receptor′s States	Ligand	Reference
A_1_AR					
5UEN	3.2	IC50: 24.9 nM [34]	Inactive	Antagonist (DU172)	Glukhova et al. [35]
5N2S	3.3	Ki: 0.7 nM [36]	Inactive	Antagonist (PSB36)	Cheng et al. [36]
6D9H	3.6	Ki: 5.10 nM [37]	Active	Agonist (adenosine)	Draper-Joyce et al. [38]
7LD3	3.2	Ki: 5.10 nM [37]	Active	Agonist (adenosine)	Draper-Joyce et al. [39]
7LD4	3.3	-	Active	Agonist and PAM (adenosine and MIPS521)	Draper-Joyce et al. [39]
A_2A_AR					
7MR5	2.8	Ki: 0.1 nM [40]	Inactive	Antagonist (ZM241385)	Martynowycz et al. [41]
7ARO	3.1	-	Inactive	Partial agonist (LUF5833)	Amelia et al. [42]
6LPL	2.0	Ki: 0.1 nM [40]	Inactive	Antagonist (ZM241385)	Ihara et al. [43]
6LPK	1.8	Ki: 0.1 nM [40]	Inactive	Antagonist (ZM241385)	Ihara et al. [43]
6LPJ	1.8	Ki: 0.1 nM [40]	Inactive	Antagonist (ZM241385)	Ihara et al. [43]
6WQA	2.0	Ki: 0.1 nM [40]	Inactive	Antagonist (ZM241385)	Lee et al. [44]
6ZDV	2.1	pKD: 5.90 [45]	Inactive	Antagonist (PubChem CID 984073)	Jespers et al. [45]
6ZDR	1.9	pKD: 8.60 [45]	Inactive	Antagonist (PubChem CID 740769)	Jespers et al. [45]
6S0L	2.7	Ki: 0.1 nM [40]	Inactive	Antagonist (ZM241385)	Nass et al. [46]
6S0Q	2.7	Ki: 0.1 nM [40]	Inactive	Antagonist (ZM241385)	Nass et al. [46]
6PS7	1.9	Ki: 0.1 nM [40]	Inactive	Antagonist (ZM241385)	Ishchenkoa et al. [47]
6JZH	2.3	Ki: 0.1 nM [40]	Inactive	Antagonist (ZM241385)	Shimazu et al. [48]
6GT3	2.0	Ki: 1.7 nM [49]	Inactive	Antagonist (AZD4635)	Borodovsky et al. [49]
6MH8	4.2	Ki: 0.1 nM [40]	Inactive	Antagonist (ZM241385)	Martin-Garcia et al. [50]
6GDG	4.1	Ki: 20 nM [51]	Active	Agonist (NECA)	García-Nafría et al. [52]
5WF5	2.6	Ki: 17.3 nM [53]	Active	Agonist (UK-432097)	White et al. [53]
5WF6	2.9	Ki: 17.3 nM [53]	Active	Agonist (UK-432097)	White et al. [53]
5OLH	2.6	pkD: 9.0 [54]	Inactive	Antagonist (Vipadenant)	Rucktooa et al. [54]
5OM1	2.1	pkD: 9.6 [54]	Inactive	Antagonist (PubChem CID 135566609)	Rucktooa et al. [54]
5OLG	1.9	Ki: 0.1 nM [40]	Inactive	Antagonist (ZM241385)	Rucktooa et al. [54]
5OLO	3.1	Ki: 0.3 nM [55]	Inactive	Antagonist (Tozadenant)	Rucktooa et al. [54]
5OM4	2.0	Ki: 1.41 nM [56]	Inactive	Antagonist (PubChem CID 135566609)	Rucktooa et al. [54]
5OLV	2.0	Ki: 5.9 nM [57]	Inactive	Antagonist (CHEMBL 1671936)	Rucktooa et al. [54]
5OLZ	1.9	Ki: 1.51 nM [56]	Inactive	Antagonist (PubChem CID 135566609)	Rucktooa et al. [54]
6AQF	2.5	Ki: 0.10 nM [40]	Inactive	Antagonist (ZM241385)	Eddy et al. [58]
5VRA	2.4	Ki: 0.10 nM [40]	Inactive	Antagonist (ZM241385)	Broecker et al. [59]
5NM2	2.0	Ki: 0.1 nM [40]	Inactive	Antagonist (ZM241385)	Weinert et al. [60]
5NM4	1.7	Ki: 0.10 nM [40]	Inactive	Antagonist (ZM241385)	Weinert et al. [60]
5NLX	2.1	Ki: 0.10 nM [40]	Inactive	Antagonist (ZM241385)	Weinert et al. [60]
5N2R	2.8	-	Inactive	Antagonist (PSB36)	Cheng et al. [36]
5MZP	2.1	Ki: 5011.87 nM [61]	Inactive	Caffeine	Cheng et al. [36]
5MZJ	2.0	Ki: 0.60 nM [62]	Inactive	Theophylline	Cheng et al. [36]
5JTB	2.8	Ki: 0.10 nM [40]	Inactive	Antagonist (ZM241385)	Melnikov et al. [62]
5UVI	3.2	Ki: 0.10 nM [40]	Inactive	Antagonist (ZM241385)	Martin-Garcia et al. [63]
5UIG	3.5	-	Inactive	Antagonist (PubChem CID 124081196)	Sun et al. [64]
5K2A	2.5	Ki: 0.10 nM [40]	Inactive	Antagonist (ZM241385)	Batyuk et al. [65]
5K2D	1.9	Ki: 0.10 nM [40]	Inactive	Antagonist (ZM241385)	Batyuk et al. [65]
5K2C	1.9	Ki: 0.10 nM [40]	Inactive	Antagonist (ZM241385)	Batyuk et.al. [65]
5K2B	2.5	Ki: 0.10 nM [40]	Inactive	Antagonist (ZM241385)	Batyuk et al. [65]
5G53	3.4	Ki: 1.00 nM [66]	Active	Agonist (NECA)	Carpenter et al. [67]
5IU4	1.7	Ki: 0.10 nM [40]	Inactive	Antagonist (ZM241385)	Segala et al. [68]
5IU8	2.0	Ki: 18 nM [68]	Inactive	Antagonist (Q27456347)	Segala et al. [68]
5IUB	2.1	Ki: 0.35 nM [68]	Inactive	Antagonist (Q27456347)	Segala et al. [68]
5IUA	2.2	Ki: 1.5 nM [68]	Inactive	Antagonist (6DX)	Segala et al. [68]
5IU7	1.9	Ki: 1.1 nM [68]	Inactive	Antagonist (6DX)	Segala et al. [68]
4UG2	2.6	Ki: 8.80 nM [69]	Active	Agonist (CGS-21680)	Lebon et al. [70]
4UHR	2.6	Ki: 8.80 nM [69]	Active	Agonist (CGS-21680)	Lebon et al. [70]
4EIY	1.8	Ki: 0.10 nM [40]	Inactive	Antagonist (ZM241385)	Liu et al. [71]
3UZC	3.3	Kd: 0.25 nM [56]	Inactive	Antagonist (PubChem CID 135566609)	Congreve et al. [56]
3UZA	3.3	Ki: 7.76 [56]	Inactive	Antagonist (PubChem CID 56844240)	Congreve et al. [56]
3VGA	3.1	Ki: 0.10 nM [40]	Inactive	Antagonist (ZM241385)	Hino et al. [72]
3VG9	2.7	Ki: 0.10 nM [40]	Inactive	Antagonist (ZM241385)	Hino et al. [72]
3PWH	3.3	Ki: 0.10 nM [40]	Inactive	Antagonist (ZM241385)	Dore et al. [73]
3RFM	3.6	Ki: 5011.87 nM [61]	Inactive	Antagonist (Caffeine)	Dore et al. [73]
3REY	3.3	Kd: 10 nM [73]	Inactive	Antagonist (XAC)	Dore et al. [73]
2YDO	3.0	Ki: 30.9 nM [74]	Active	Agonist (adenosine)	Lebon et al. [56]
2YDV	2.6	Ki: 13.8 nM [74]	Active	Agonist (NECA)	Lebon et al. [56]
3QAK	2.7	Ki: 4.75 nM [75]	Active	Agonist (UK-432097)	Xu et al. [75]
3EML	2.6	Ki: 0.10 nM [40]	Inactive	Antagonist (ZM241385)	Jaakola et al. [76]

## Data Availability

Not applicable.

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
