# Peer review of "Molecular Simulations and Drug Discovery of Adenosine Receptors"

_molecules, 2022, doi:10.3390/molecules27072054_

Round 1

Reviewer 1 Report

This review is well designed and could be accepted in the current form.

Reviewer 2 Report

The aim of this paper was to  summarized the most recent data about molecular simulation of adenosine receptors and drug targets for a possible teratment of human disease in the area of inflammation, cardiac ischemia, stropke, pain and cancer.

The paer is well writen and of great interest

I have some minor concernes

  1. Introduction

While in summary the authors  introduce the role of adenosine receptor targeting in myocardial ischemia, this not appears in the text and specially in Introduction

Recent revue on this subject should be helpful Paganelli F et al Cardiovasc Res 2021

 In the area of inflammation, Eltzschog HK group has propably the best experience and should be cited Eltzschig HK and Sitkovski Purinergic signaling during Inflammation

New Eng J Med 2011

2.1

Line 5 : Ref 27  (Carpenter ) and 28 (Garcia-Nafria)  concern A2A receptors and not A1.

2.3  The first paragraph may be improved..

The term biased agonism was first intriduced by Jarpe et al 1998 J Biol Chem 273 :3097

About subtance P and concern a phenomenom  by which a ligand preferentially  activates one signaling pathway among several. This is an important notion because  this could lead to the synthesis of drugs without adverse effects.

2.5 Page 14

 In another study [77], Degunati et al should be replaced by In another study, Degunati et al [77]….to harmonize the ref

2.6 Lipid interactions

Very well written

It is not precised if cholesterol or other lipids  may alter the binding of ligand for A1 or A2A receptors in term of affinity and or EC50 at the extracellular level and if PIP2 may alter via flexibility alteration, the transduction signal pathway?

This is important for the development of new drugs on one hand and if as example hypercholetrolmia in human may alter the binding of ADO to A2A R?

There is  a lot of abbreviations

It should be interesting to have a special page with the main abbreviations.

But overall nice work

Reviewer 3 Report

The review article “Molecular simulations and drug discovery of adenosine receptors” by Wang et. al. provides a review on recent studies in molecular simulations and computer-aided drug discovery of the adenosine receptors.

The following concerns must be addressed prior to acceptance of the paper.

  1. In panel A in Figure 5, the traces seem to overlap, please consider showing the more visible depiction of traces.
  2. In Table 1, please consider adding the available binding affinity data for each entry e.g., IC50.
  3. In section 1, please consider briefly describing each adenosine receptor separately.
  4. In section 1, please consider adding the schematic figure for depicting the adenosine signaling.
  5. In section 2.4, please consider showing the figure depicting the allosteric binding site.
  6. In section 3, please briefly clarify how MD simulations have been incorporated into the virtual screening.
  7. In section 3.1, the authors should cover other free energy calculation methods beyond MD/FEP, e.g., several adenosine receptor drug discovery articles report MM-PBSA or MM-GBSA for free-energy calculation.
  8. In section 3.3, the authors should improve this section, they should describe the induced-fit docking for conducting flexible docking, or other methods for extracting structural ensembles such as NMR ensembles.
  9. In section 4, the authors should briefly expand their description of advanced computational platforms, e.g., there are articles reporting using deep learning platforms for drug discovery of adenosine receptors.

Round 2

Reviewer 3 Report

While the implemented changes are appreciated, panel A in Figure 5 is still vague, the traces are overlapping with each other significantly. Authors should redesign Panel A such that the distance traces can clearly be compared with each other. Authors may simply divide the traces as separate plots or smooth the traces using any standard plotting software.
